# Proliferative Vitreoretinopathy in Retinal Detachment: Perspectives on Building a Digital Twin Model Using Nintedanib

**DOI:** 10.3390/ijms252011074

**Published:** 2024-10-15

**Authors:** Giacomo Visioli, Annalisa Romaniello, Leonardo Spinoglio, Giuseppe Maria Albanese, Ludovico Iannetti, Oscar Matteo Gagliardi, Alessandro Lambiase, Magda Gharbiya

**Affiliations:** 1Department of Sense Organs, Medicine and Dentistry Faculty, Sapienza University of Rome, Viale del Policlinico 155, 00161 Rome, Italy; giacomo.visioli@uniroma1.it (G.V.); annalisa.romaniello@uniroma1.it (A.R.); giuseppemaria.albanese@uniroma1.it (G.M.A.); oscarmatteo.gagliardi@uniroma1.it (O.M.G.); alessandro.lambiase@uniroma1.it (A.L.); 2Ophthalmology Unit, Head and Neck Department, Policlinico Umberto I University Hospital, 00161 Rome, Italy; ludovicoiannetti@gmail.com

**Keywords:** proliferative vitreoretinopathy, computational models, digital twin, nintedanib, fibrosis, retinal detachment, signaling pathways, personalized medicine, ophthalmology

## Abstract

Proliferative vitreoretinopathy (PVR) is a pathological process characterized by the formation of fibrotic membranes that contract and lead to recurrent retinal detachment. Pars plana vitrectomy (PPV) is the primary treatment, but recurrence rates remain high, as surgery does not address the underlying molecular mechanisms driving fibrosis. Despite several proposed pharmacological interventions, no approved therapies exist, partly due to challenges in conducting preclinical and in vivo studies for ethical and safety reasons. This review explores the potential of computational models and Digital Twins, which are increasingly gaining attention in medicine. These tools could enable the development of progressively complex PVR models, from basic simulations to patient-specific Digital Twins. Nintedanib, a tyrosine kinase inhibitor targeting PDGFR, VEGFR, and FGFR, is presented as a prototype for computational models to simulate its effects on fibrotic pathways in virtual patient cohorts. Although still in its early stages, the integration of computational models and Digital Twins offers promising avenues for improving PVR management through more personalized therapeutic strategies.

## 1. Introduction

Proliferative vitreoretinopathy (PVR) remains a significant complication following retinal detachment (RD), leading to recurrent detachment in a considerable percentage of cases, even after successful surgical repair [1,2]. PVR is characterized by the formation of epiretinal and subretinal fibrotic membranes, which apply tractional forces on the retina, often resulting in refractory or recurrent retinal detachment [2,3]. Currently, the only treatment for PVR is surgical, typically via pars plana vitrectomy (PPV), aimed at removing the vitreous gel, excising fibrotic membranes, and stabilizing the retina. Despite advancements in surgical techniques, PVR continues to have high recurrence rates, and visual recovery is often limited due to the fibrotic damage caused by the disease [4].

To date, no pharmacological therapies have been approved for the treatment of PVR [5]. However, various preclinical studies have explored the potential of pharmacological agents to complement surgical interventions in preventing PVR recurrence [6,7]. Molecules targeting key pathways involved in PVR, such as inhibitors of TGF-β, PDGF, and VEGF, have been tested in vitro and in animal models [8,9]. Despite these promising findings, challenges remain in translating these treatments into clinical practice due to limited in vivo studies and concerns surrounding safety and efficacy in humans.

Given these limitations, there is a growing interest in computational models as a tool for studying the behavior of complex signaling pathways [10,11]. Computational models have the potential to simulate molecular interactions and the effects of pharmacological agents without the need for invasive procedures. These models have been applied successfully in other areas of medicine to predict disease progression and treatment outcomes [12]. However, in ophthalmology, including PVR, their application remains largely unexplored [13].

While simulation models represent a predefined, static, or time-limited scenario based on initial assumptions, describing a dynamic system like the human body requires an evolving approach. One emerging concept in this area is the Digital Twin (DT)—a virtual reproduction of a real system or process that allows for testing, monitoring, and simulation of the physical counterpart [14]. Unlike simulation models operating on fixed parameters, DTs provide real-time digital replicas that adapt to patient-specific data. This allows them to track individuals’ specific progression and response to events. Incorporating statistical models into DT frameworks could enhance the context of predictions, improving both diagnostic accuracy and personalized treatment strategies [15].

In healthcare, DTs are being developed to model organs, diseases, and personalized treatment strategies [16,17]. In the context of PVR, a DT could simulate the effects of various pharmacological agents, such as Nintedanib, on key signaling pathways, including PDGFR, VEGFR, and FGFR, which play a critical role in epithelial–mesenchymal transition (EMT) and fibrotic processes [18]. By virtually testing these interventions, a DT could help predict their efficacy, optimize treatment strategies, and reduce the need for invasive in vivo testing.

Although computational models and Digital Twin applications are not yet widely used or integrated into clinical practice for ophthalmological diseases, the potential for developing such models is significant [19]. This review will examine the literature on PVR, highlighting key studies and exploring how these findings can guide the development of future computational models. These models could simulate the progression of PVR and assess the impact of treatments like Nintedanib on key signaling pathways. By doing so, we aim to provide a foundation for optimizing therapeutic strategies in retinal detachment surgery.

## 2. Proliferative Vitreoretinopathy: Epidemiology and Risk Factors

PVR is a serious complication of rhegmatogenous retinal detachment (RD), affecting between 5.1% and 11.7% of cases. PVR can significantly hinder successful retinal reattachment due to the formation of fibrotic membranes, which cause tractional forces on the retina [20,21]. These forces can eventually lead to tractional retinal detachment, a more challenging condition to manage surgically. Despite advances in vitreoretinal surgery, anatomical success rates for RD complicated by PVR range from 10% to 40%, even after multiple interventions [22].

The development of PVR typically occurs within the first six weeks after retinal detachment and is often exacerbated by intraoperative factors, such as vitreous hemorrhage, cryotherapy, photocoagulation, or tamponade [4]. Delaying surgical intervention may also accelerate PVR progression, as the condition becomes more complex with time. The presence of multiple retinal breaks and extensive retinal detachments increases the risk. Genetic predispositions, such as polymorphisms in TNF, IL1A, MDM2, and p53, have also been linked to a heightened susceptibility to PVR [23,24]. Additionally, polymorphisms in the apoptosis regulators BAX and BCL-2 have been associated with an increased risk of PVR [24]. Lifestyle factors, such as smoking, are thought to aggravate the condition, further complicating outcomes post-surgery [25,26].

### 2.1. Pathophysiology of PVR

The pathogenesis of PVR is complex, involving various molecular and cellular processes that lead to the formation of fibrotic membranes and retinal contraction [27]. A key mechanism in the development of PVR is the EMT of retinal pigment epithelial (RPE) cells. During EMT, RPE cells transform into myofibroblasts that secrete extracellular matrix (ECM) proteins, contributing to the formation of fibrotic membranes [28,29]. These membranes exert tractional forces on the retina, often resulting in recurrent detachment.

Several cytokines and growth factors are heavily implicated in this process, including transforming growth factor β (TGF-β), platelet-derived growth factor (PDGF), vascular endothelial growth factor (VEGF), fibroblast growth factor (FGF), and tumor necrosis factor-α (TNF-α) [30,31,32,33,34,35]. These molecules activate intracellular signaling pathways that regulate cell proliferation, migration, and differentiation, all of which contribute to the fibrotic process characteristic of PVR [36,37,38].

The progression of PVR typically begins with two main events: the breakdown of the blood–retinal barrier and hypoxia, which leads to photoreceptor death [39]. In response to these events, RPE cells, along with glial cells, macrophages, and lymphocytes, are activated and undergo the EMT process, transforming into mesenchymal cells that produce fibrotic membranes [40]. Pigment epithelium-derived factor (PEDF) has also been identified as a key regulator in this complex process, influencing cell survival and the formation of fibrotic tissue [41,42]. In terms of biomarkers, EMT in PVR is associated with an increased expression of fibronectin, N-cadherin, alpha-smooth muscle actin (α-SMA), and vimentin, along with a decreased expression of E-cadherin and zona occludens-1 (ZO-1) [43,44,45,46]. These biomarkers reflect the transition of epithelial cells to a mesenchymal state, and their regulation is crucial to the fibrotic response in PVR. A schematic representation of PVR development is reported in Figure 1.

More recently, a newer concept of the pathogenesis of PVR has emerged, suggesting that retinal glial cells play a critical role in the PVR healing response, particularly following mechanical injury to the inner retina caused by posterior vitreous detachment (PVD). It is proposed that the shearing forces generated during PVD can cause subclinical damage to the inner retina, activating glial cells [47].

### 2.2. Classification of PVR

The Retina Society updated the classification system for PVR in 1991, refining the original scheme to better describe the location and severity of fibrotic proliferation [48]. The system is divided into Grades A, B, and C, with the more severe Grade C being subdivided into C-P (posterior to the equator) and C-A (anterior to the equator). Each of these is further divided into types, based on the extent and characteristics of the retinal folds and subretinal strands, as reported in Table 1.

Despite this detailed classification, it is rarely used in routine clinical practice due to its complexity [49]. However, with further research into the pathogenesis of PVR, particularly in relation to the timing of interventions, this classification could be a useful tool in identifying the optimal stage for pharmacological intervention.

### 2.3. Surgical Approach

In the management of PVR, complete removal of the vitreous is essential and is achieved via PPV, followed by meticulous shaving of the vitreous base with scleral depression assistance [50]. Peeling of fibrotic membranes is often facilitated using magnification techniques and dyes, such as trypan blue or brilliant blue, to enhance visualization of the membranes [51,52,53]. Chandelier endoillumination has proven especially useful in advanced cases, as it enables a bimanual approach for more precise dissection and membrane removal [54,55].

In advanced cases of PVR, particularly recurrent detachments, retinectomy plays a crucial role in the surgical approach [56]. Retinectomy involves the removal of portions of the retina that have become fibrotic or rigid, preventing proper retinal reattachment. Partial retinectomies are frequently performed to relieve retinal traction and facilitate the flattening of the retina. In the most severe cases, a 360-degree retinectomy may be necessary, leaving only the posterior pole of the retina intact. This extensive retinectomy is used when intraretinal stiffness makes it impossible to achieve retinal reattachment using other methods. Silicone oil (SiO) tamponade is often employed following retinectomy to help stabilize the remaining retina and prevent further detachment [57]. Scleral buckling may sometimes be combined with PPV to alleviate tractional forces and improve retinal reattachment outcomes [58].

More recently, the use of amniotic membrane grafts during vitrectomy has emerged as a promising adjunct therapy [59]. By covering the RPE, the amniotic membrane can suppress the TGF-β pathway, limiting fibrosis by acting on fibroblasts, macrophages, and RPE cells [60]. This strategy has shown potential in reducing PVR recurrence and improving postoperative outcomes, particularly in recurrent or complex cases. Despite these advances, even in the hands of the most skilled surgeons utilizing the latest surgical techniques, recurrence rates remain high, with poor long-term visual outcomes and retinal anatomical integrity [61]. This has led to increasing interest in combining surgical intervention with pharmacological approaches to enhance treatment efficacy and reduce the risk of PVR recurrence.

## 3. Research on Pharmacological Treatment

Parallel to the surgical solution, research has focused on identifying predictive biomarkers that can help determine the risk of PVR development and guide individualized treatment strategies [62,63]. More importantly, there has been increasing interest in pharmacological approaches to managing PVR, with the goal of combining surgery with pre- and post-operative drug therapies to reduce PVR formation and recurrence [64]. The idea is that targeting key signaling pathways associated with PVR pathogenesis could minimize fibrotic membrane formation, ultimately improving surgical outcomes. A major focus has been on intravitreal drug delivery, with various therapeutic agents undergoing testing in vitro and in vivo, primarily in experimental animal models. Among the first to be explored were intravitreal implants of dexamethasone and triamcinolone acetonide, used both preventively and post-operatively to reduce inflammation and fibrosis [65,66,67].

Other pharmacological approaches have targeted antiproliferative pathways, with molecules such as 5-fluorouracil (5-FU), retinoic acid, daunorubicin, Taxol, ribozymes, vincristine, cisplatin, Adriamycin, and mitomycin showing varying degrees of success in preclinical models [5,7,68,69]. Anti-VEGF agents have also been explored for their potential to limit vascular proliferation and reduce membrane contraction in PVR, though the results are mixed and inconclusive [70].

The off-label use of intravitreal methotrexate (MTX) during PPV for advanced PVR has not demonstrated significant benefits compared to PPV alone regarding retinal reattachment, but it has shown potential for safety and functional outcomes in preventing mild PVR, as noted in a study by Ullah et al. [71]. Further research has also highlighted promising results from molecules such as resveratrol, epigallocatechin gallate, N-acetylcysteine, 5-FU, and low-molecular-weight heparin in improving the success of pars plana vitrectomy (PPV) in cases of PVR following retinal detachment [71,72,73,74]. These studies suggest that pharmacological strategies aimed at interrupting the signaling pathways involved in PVR could serve as valuable adjuncts to surgical treatment, reducing the likelihood of recurrence and improving long-term outcomes.

### 3.1. Drug Delivery Issues

Drug delivery to the vitreous chamber poses significant challenges, particularly regarding the safety and toxicity of pharmacological agents when administered directly into this sensitive environment. The vitreous chamber is delicate, and improper delivery or drug accumulation can result in toxic effects on the retina or adjacent structures, potentially leading to adverse outcomes, such as retinal damage, inflammation, or increased intraocular pressure.

A major concern in intravitreal drug delivery is ensuring that the drug remains within the vitreous for an adequate duration to exert its therapeutic effects. Many drugs are cleared from the vitreous relatively quickly, requiring frequent injections, not only increasing the risk of complications but also reducing patient compliance. To address this, researchers have explored various slow-release devices that allow for prolonged and controlled drug release, minimizing the need for repeated interventions [75,76].

For example, Xiao et al. demonstrated in vitro that the combination of dexamethasone and daunorubicin has enhanced anti-PVR effects compared to the use of either drug alone [77]. These drugs were delivered using oxidized porous silicon particles, showing promising results in terms of inhibiting contraction and proliferation, key processes in PVR progression. This delivery system allowed for sustained drug release within the vitreous. Further studies by Warther et al. [78] confirmed that the drugs could persist in the vitreous for up to three months, while Nieto et al. demonstrated that the oxidized porous silicon particles could safely degrade within the vitreous, reducing concerns about long-term toxicity [79]. In addition to porous silicon particles, other slow-release devices have shown potential in PVR treatment. Lipo-dox^®^, a liquid suspension containing doxorubicin encapsulated in unilamellar hydrochloride liposomes, has demonstrated promising results in experimental models of PVR [80]. This formulation allows for the slow and stable release of doxorubicin into the vitreous, ensuring prolonged therapeutic effects. A schematic representation of the drug-release profiles comparing standard intravitreal injection to other delivery devices is shown in Figure 2.

### 3.2. Nintedanib as a New Therapeutic Option

Nintedanib is a multi-target tyrosine kinase inhibitor currently approved for the treatment of idiopathic pulmonary fibrosis (IPF). It functions by blocking the activation of several receptor tyrosine kinases (RTKs), including platelet-derived growth factor receptors (PDGFRs) α and β, fibroblast growth factor receptors (FGFRs) 1–3, and vascular endothelial growth factor receptors (VEGFRs) 1–3 [81,82]. These kinases play crucial roles in processes such as cell proliferation, migration, and angiogenesis, which are key in both normal tissue repair and pathological fibrosis. In addition to its effects on RTKs, Nintedanib also inhibits non-receptor tyrosine kinases (NRTKs), like Lck, Lyn, and Src, as well as the colony-stimulating factor 1 receptor (CSF1R) [83]. By competitively binding to the ATP-binding site of these kinases, Nintedanib interrupts intracellular signaling pathways that contribute to the development of fibrosis. This mechanism has been demonstrated to effectively prevent the transformation of primary human lung fibroblasts into myofibroblasts when induced by TGF-β2, showing significant inhibition of fibroblast proliferation, migration, and extracellular matrix deposition [84,85]. A diagram representing the functions of Nintedanib is shown in Figure 3.

Given these antifibrotic properties, Nintedanib has been investigated for its potential application in ocular fibrosis. In vitro studies have shown that Nintedanib can inhibit the epithelial–mesenchymal transition (EMT) of retinal pigment epithelial (RPE) cells induced by TGF-β2, which may help reduce the formation of fibrotic membranes in conditions like proliferative vitreoretinopathy (PVR) [86]. Specifically, Nintedanib was found to prevent the reduction of E-cadherin expression and the upregulation of mesenchymal markers such as fibronectin, N-cadherin, and α-SMA, key indicators of EMT [87]. In cell culture models using ARPE-19 cells, Nintedanib demonstrated dose-dependent inhibition of TGF-β2-induced cell proliferation, migration, and contraction, thereby suggesting a potential role in reducing fibrotic membrane formation in the eye [18]. While the in vitro studies on Nintedanib are promising, showing its potential to inhibit key processes involved in ocular fibrosis, the differences between laboratory models and native retinal cells should be considered. For instance, RPE cells used in these studies often exhibit characteristics more similar to diseased or aged cells, such as weaker tight junctions and less pigmentation [88]. These differences may affect how Nintedanib performs in a real clinical setting, suggesting the need for further validation in animal models. Further investigation into its potential in ocular fibrosis, particularly in clinically relevant models, is necessary to support its application in human trials. With this in mind, we can explore whether virtual models of PVR could help bridge this gap, simulating animal models and testing various conditions in a manner that mimics real-world scenarios.

## 4. Can Computational Models and Digital Twins Aid in Treating PVR?

Traditionally, in vivo testing, supported by in vitro studies, has been the gold standard for evaluating the effects of pharmacological agents in PVR [89]. However, with the growing number of potential treatments, these approaches have become increasingly time-consuming and costly, especially in cases where even in vitro or animal studies present logistical challenges. To address these limitations, computational models and DTs could offer a new frontier for simulating and testing drug effects on pathological processes in a more efficient and cost-effective manner [14,90].

Computational models are algorithmic frameworks designed to simulate biological processes, allowing researchers to test hypotheses and predict how various interventions might influence disease progression. DTs extend this concept by creating dynamic virtual representations of patients, organs, or disease states. Indeed, DTs are gaining significant traction across various fields of medicine due to their capacity to model complex systems and provide personalized insights. Initially developed for industrial applications, DTs have increasingly been adapted to healthcare, where their use is rapidly expanding. In clinical contexts, DTs are being developed to simulate individual patient conditions, predict disease progression, and optimize treatment strategies [16]. For example, DTs have been successfully applied in cardiology to create virtual heart models that help predict outcomes of surgical interventions or monitor heart disease progression. Similarly, in oncology, DTs are being used to simulate tumor growth and predict how different cancer therapies might affect individual patients, enabling more personalized and precise treatment plans [91,92]. As the technology evolves, DTs are becoming increasingly sophisticated, not only modeling individual organs but also representing entire systems; complex disease processes; or even broader healthcare scenarios, such as prevention strategies. For instance, in diabetes care, DTs are being developed to monitor and predict glucose fluctuations, allowing for real-time adjustments to treatment regimens [93]. In neurology, researchers are exploring the use of DTs to simulate neurodegenerative diseases, offering a virtual platform to test interventions that may slow disease progression [94].

Although not yet explored, DTs could have a potential role in ophthalmology, both in anterior and posterior segment diseases [16]. They could be instrumental in predicting clinical outcomes, particularly for complex or tailored treatments. For example, in anterior segment conditions like keratoconus or glaucoma, DTs could simulate disease progression and optimize surgical interventions. Similarly, in posterior segment diseases such as age-related macular degeneration and diabetic retinopathy, DTs could offer more personalized treatment approaches [95]. In rare ocular diseases like retinitis pigmentosa, where patient populations are small and treatment options are limited, DTs could be used to predict disease progression and treatment responses, offering a valuable tool in rare disease management. Furthermore, DTs could be employed to test surgical outcomes; choose the best technique for a specific eye; or be used for educational purposes, such as surgical training [96].

In the case of PVR, a DT could be constructed by generating a virtual cohort based on real patient data [97]. This model would allow for the manipulation of factors such as cytokine levels, genetic predispositions, or surgical outcomes, providing a platform to test drug combinations and predict individualized responses. Additionally, DTs offer the potential to reduce the ethical concerns of placebo-controlled trials by creating synthetic control groups using actual patient data, thus expediting drug development while minimizing patient risk [98].

The integration of DTs into PVR research could ultimately revolutionize our understanding of this complex condition. By simulating drug–disease interactions in a virtual environment, researchers can explore a broader range of therapeutic possibilities, accelerating the development of novel treatments and enabling more personalized care, ultimately improving patient outcomes.

### Nintedanib as Prototype of a Computational Model

Before developing advanced Digital Twins (DTs), it is essential to start with simpler computational models and progressively build more complex systems. These models serve as the foundation for constructing representations that could eventually link to real clinical cases, where the digital model might function as a “twin” of the actual patient’s condition.

Nintedanib represents a compelling example of a therapeutic agent to be tested in an early-stage computational model, as it inhibits multiple signaling pathways involved in disease progression [87]. In Figure 4, we illustrate a theoretical example we generated using a basic computational simulation developed in Python (Version 3.12.5) to explore how Nintedanib could affect receptor inhibition and its potential impact on cellular fibrosis [99]. The simulation aims to represent the hypothetical dynamics of receptor activation (e.g., PDGFR, VEGFR, and FGFR) and suggests how their inhibition, following Nintedanib administration, might influence fibrosis levels over time. This conceptual framework is used to demonstrate the potential utility of computational models in visualizing and exploring therapeutic mechanisms, emphasizing that adjusting variables in the system could theoretically lead to different outcomes for receptor inhibition and fibrosis progression.

Beyond this illustrative example, it is important to note that this model represents just one possible scenario. By generating a virtual cohort of patients with varying stages and values of PVR, different outcomes could emerge depending on each patient’s specific characteristics. This demonstrates the potential of computational models to account for patient variability and illustrates how future research could aim to develop more individualized predictions.

This is only the starting point. The model can be further refined by incorporating additional molecular complexities and integrating more signaling pathways, such as the inhibition of TGF-β, CSF1R, Lyn, and Lck, as well as non-inhibited factors like cytokines (e.g., TNF-α, IL-1, IL-6, and IL-8) [81]. By progressively expanding these molecular and clinical inputs, the model can evolve into a more comprehensive tool, moving closer to the goal of creating a DT that could predict patient-specific responses to therapy and improve treatment strategies for PVR.

## 5. Conclusions

Proliferative vitreoretinopathy (PVR) remains a significant complication following retinal detachment, often leading to recurrent detachment despite advances in surgical techniques. Currently, no pharmacological therapies are approved to prevent PVR recurrence, and translating promising preclinical findings into clinical practice is challenging due to limited in vivo data and safety concerns.

In this article, we simulated the potential outcomes of a computational model for Nintedanib, illustrating its effects on key signaling pathways involved in PVR. By modeling receptor inhibition and its impact on fibrosis, we demonstrated how such models could help explore therapeutic mechanisms. Additionally, by simulating a virtual patient cohort, we showcased how individualized responses to treatment could be predicted, laying the groundwork for more personalized approaches.

While computational models and Digital Twins are still in the pioneering phase within ophthalmology, their potential to simulate disease processes and optimize treatment strategies is significant. With further development, these tools could transform how PVR is managed, offering tailored treatments and improving patient outcomes.

## Figures and Tables

**Figure 1 ijms-25-11074-f001:**
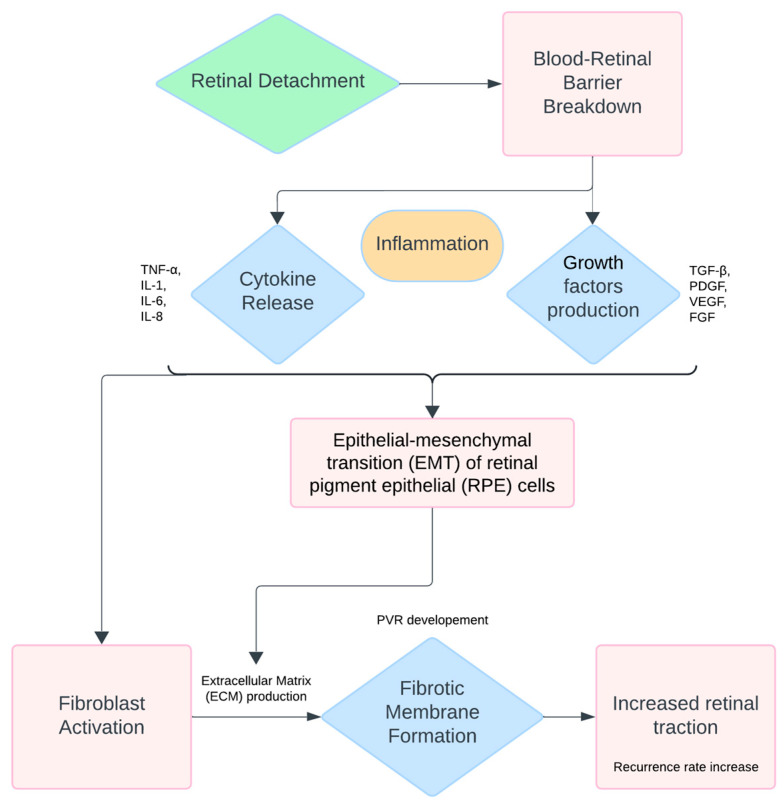
Pathogenesis of proliferative vitreoretinopathy (PVR), showing the progression from retinal detachment to blood–retinal barrier breakdown, inflammation, and the activation of cytokines and growth factors, leading to epithelial–mesenchymal transition, fibrotic membrane formation, and retinal contraction.

**Figure 2 ijms-25-11074-f002:**
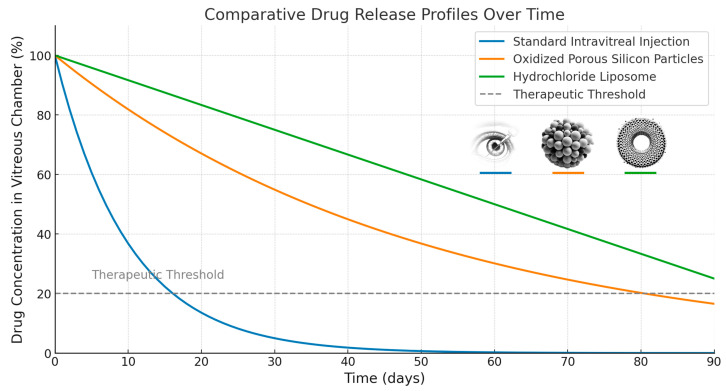
Schematic rendering of drug-release profiles in the vitreous chamber for standard intravitreal injection, oxidized porous silicon particles, and hydrochloride liposomes, compared to the therapeutic threshold.

**Figure 3 ijms-25-11074-f003:**
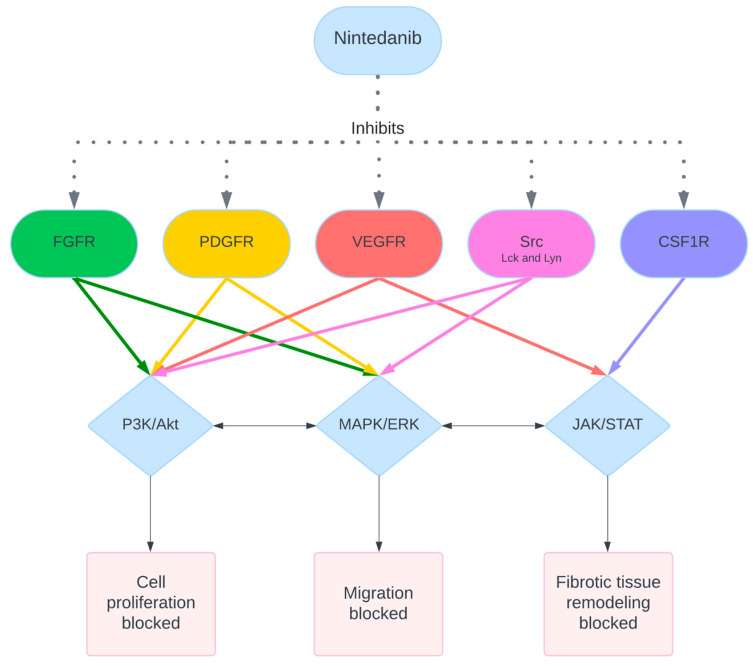
Schematic representation of the signaling pathways blocked by Nintedanib. Nintedanib inhibits FGFR (fibroblast growth factor receptor), PDGFR (platelet-derived growth factor receptor), VEGFR (vascular endothelial growth factor receptor), Src kinases (Lck and Lyn), and CSF1R (colony-stimulating factor 1 receptor). This inhibition affects the downstream signaling pathways, including PI3K/Akt, MAPK/ERK, and JAK/STAT, resulting in the blockage of cell proliferation, migration, and fibrotic tissue remodeling. Dashed arrows, directional arrows, and bidirectional arrows represent contemporaneous, direct, and indirect effects, respectively.

**Figure 4 ijms-25-11074-f004:**
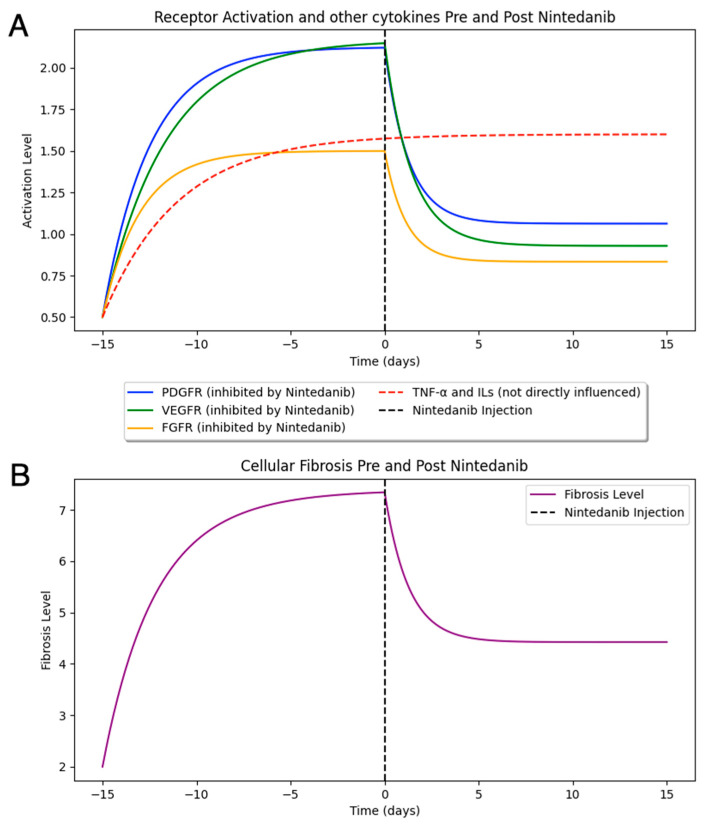
Nintedanib inhibits key receptors involved in fibrotic signaling pathways: PDGFR (platelet-derived growth factor receptor), VEGFR (vascular endothelial growth factor receptor), and FGFR (fibroblast growth factor receptor), while TNF-α (tumor necrosis factor-α) and some interleukins (ILs) are not directly influenced. The reduction in receptor activation levels post-administration (**A**) leads to a decrease in cellular fibrosis over time (**B**). This figure illustrates a possible output from a rudimentary computational model, providing a conceptual example of how receptor inhibition induced by Nintedanib might impact fibrosis progression.

**Table 1 ijms-25-11074-t001:** Classification of proliferative vitreoretinopathy according to the Retina Society.

Grade and Type	Clinical Signs
A	Vitreous haze, pigment clumps, and pigment clusters on inferior retina.
B	Wrinkling of inner retinal surface, retinal stiffness, vessel tortuosity, rolled and irregular edge of retinal break, and decreased mobility of vitreous.
C-P (posterior)	Full-thickness retinal folds or subretinal strands posterior to equator (1–12 clock hours involvement).
Type	(i) Starfolds posterior to vitreous base; (ii) confluent starfolds possibly obscuring the optic disc; and (iii) subretinal proliferation with annular and linear strands, and moth-eaten-appearing sheets.
C-A (anterior)	Full-thickness retinal folds or subretinal strands anterior to equator, anterior displacement, and condensed vitreous strands.
Type	(i) Inward retina contraction at the vitreous base, with central displacement, stretched peripheral retina, and radial folds in the posterior retina; and (ii) anterior contraction at the vitreous base, with ciliary body detachment, epiciliary membrane, and iris retraction.

## Data Availability

No new data were created or analyzed in this study.

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
