# Peer review of "Proliferative Vitreoretinopathy in Retinal Detachment: Perspectives on Building a Digital Twin Model Using Nintedanib"

_ijms, 2024, doi:10.3390/ijms252011074_

Round 1
Reviewer 1 Report
Comments and Suggestions for Authors
Dear authors,
I appreciate the opportunity to review this interesting manuscript on the Integration of computational models and digital twins for PVR management. I have so far, no major concerns on the concepts and facts described. However, the introduction and facts covered on the digital twins is very short and need to be more elaborated. My comments below:
Is there any difference between digital twin (DT) and simulation model? The authors need to mention this.
Apart from PVR, what other eye/retinal diseases can be diagnosed through DT. Since this is a review paper, it would be good to mention.
Is Figure 3 represented in the review is from some paper or is your own lab study. The authors need to mention or reference this. Also, the Y-axis detail is missing in upper panel of figure 3.
If possible, to put some figures related to section 3.1 drug delivery issues, that will be great.
Author Response
Dear authors,
I appreciate the opportunity to review this interesting manuscript on the Integration of computational models and digital twins for PVR management. I have so far, no major concerns on the concepts and facts described. However, the introduction and facts covered on the digital twins is very short and need to be more elaborated.
Reply: Thank you so much for taking the time to review our article and for your encouraging comments. As suggested, we have expanded the the introduction and the sections about digital twins.
My comments below:
Is there any difference between digital twin (DT) and simulation model? The authors need to mention this.
Reply: Thank you for your question. We added a paragraph to explain the difference for a general audience in the introduction.
Apart from PVR, what other eye/retinal diseases can be diagnosed through DT. Since this is a review paper, it would be good to mention.
Reply: Thank you for raising this point. Indeed, PVR is just an example of eye disease that can be studied with DTs. So we added a paragraph in the fourth section.
Is Figure 3 represented in the review is from some paper or is your own lab study. The authors need to mention or reference this.
Reply: Figure 3 (now Figure 4) is an original elaboration that we produced for this article. We added a little specification.
Also, the Y-axis detail is missing in upper panel of figure 3.
Reply: Thank you. The image has been corrected and replaced.
If possible, to put some figures related to section 3.1 drug delivery issues, that will be great.
Reply: Thank you for your idea. Of course, we've added a new figure for that section (now referenced as Figure 2).
Reviewer 2 Report
Comments and Suggestions for Authors
This is a really unique and interesting study. Management of PVR is one of the last and core difficult problem in vitreoretinal surgery. Any strategy to overcome the pathophysiology should be highly evaluated.
I would like to comment as an retinal surgeon.
Major comment.
In the manuscript, description of Nintedanib for clinical or animal model is too limited. Since many readers are not always familiar with the agent, author should explain as much as a section.
Minor points.
page 1. line 31 "tractional retinal detachment" → "refractory retinal detachment" or "tractional or recurrence of rhegmatogenous retinal detachment"
Page 3 figure " tractional retinal detachment" should be changed to "Increased retinal traction" , since "tractional retinal detachment" usually means RD without retinal break (PDR etc).
Author Response
This is a really unique and interesting study. Management of PVR is one of the last and core difficult problem in vitreoretinal surgery. Any strategy to overcome the pathophysiology should be highly evaluated. I would like to comment as an retinal surgeon.
Reply: Thank you for taking the time to read an evaluate our article. We really appreciate that this is read by a vitreoretinal surgeon.
Major comment.
In the manuscript, description of Nintedanib for clinical or animal model is too limited. Since many readers are not always familiar with the agent, author should explain as much as a section.
Reply: Thank you for your comment. We agree that many concepts are introduced, which may be challenging for a general audience to follow. Based on your suggestion, we have simplified and expanded Section 3.1 (Nintedanib as a new therapeutic option) and provided more detail on clinical and animal models. Unfortunately, the current literature remains quite limited.
Minor points.
page 1. line 31 "tractional retinal detachment" → "refractory retinal detachment" or "tractional or recurrence of rhegmatogenous retinal detachment"
Reply: Amended. We chose the second option.
Page 3 figure " tractional retinal detachment" should be changed to "Increased retinal traction" , since "tractional retinal detachment" usually means RD without retinal break (PDR etc).
Reply: Amended. Please see the updated figure (now referenced as Figure 3).